# Imitation Learning of Robot Policies using Language, Vision and Motion

## Abstract

In this work we propose a novel end-to-end imitation learning approach which combines natural language, vision, and motion information to produce an abstract representation of a task, which is in turn used to synthesize specific motion controllers at run-time. This multimodal approach enables generalization to a wide variety of environmental conditions and allows an end-user to direct a robot policy through verbal communication. We empirically validate our approach with an extensive set of simulations and show that it achieves a high task success rate over a variety of conditions while remaining amenable to probabilistic interpretability.

## 1 Introduction

A significant challenge when designing robots to operate in the real world lies in the generation of control policies that can adapt to changing environments. Programming such policies is a labor and time-consuming process which requires substantial technical expertise. Imitation learning (Schaal, 1999), is an appealing methodology that aims at overcoming this challenge – instead of complex programming, the user only provides a set of demonstrations of the intended behavior. These demonstrations are consequently distilled into a robot control policy by learning appropriate parameter settings of the controller. Popular approaches to imitation, such as Dynamic Motor Primitives (DMPs) (Ijspeert et al., 2013) or Gaussian Mixture Regression (GMR) (Calinon, 2009) largely focus on motion as the sole input and output modality, i.e., joint angles, forces or positions. Critical semantic and visual information regarding the task, such as the appearance of the target object or the type of task performed, is not taken into account during training and reproduction. The result is often a limited generalization capability which largely revolves around adaptation to changes in the object position. While imitation learning has been successfully applied to a wide range of tasks including table-tennis Mülling et al. (2013), locomotion Chalodhorn et al. (2007), and human-robot interaction Amor et al. (2014) an important question is how to incorporate language and vision into a differentiable end-to-end system for complex robot control.

In this paper, we present an imitation learning approach that combines language, vision, and motion in order to synthesize natural language-conditioned control policies that have strong generalization capabilities while also capturing the semantics of the task. The main rationale of our approach is that a teacher typically provides substantially more information than just the kind of motion to perform. Imagine an athletic trainer that is demonstrating a tennis swing while also verbally explaining the involved steps, the target position, or the speed. As a result of this rich collection of information, the student can develop complex associations between (a) the observed visual features, (b) the demonstrated arm movement, and (c) the provided verbal descriptions. We argue that such a multi-modal teaching approach enables robots to acquire complex policies that generalize to a wide variety of environmental conditions. To this end, we propose a neural network architecture, including several sub-networks, that can be trained in an end-to-end fashion to capture the complex relationships between language, vision, and motion observed in the demonstrations. After training, the network can be provided with a camera image of the current environment and a natural language description of the intended task. The description typically corresponds to verbal commands given by the current user. In turn, the network produces control parameters for a lower-level control policy that can be run on a robot to synthesize the corresponding motion. The hierarchical nature of our approach, i.e., a high-level policy generating the parameters of a lower-level policy, allows for generalization of the trained task to a variety of spatial, visual and contextual changes. Further, the ability to provide

commands and instructions to the policy enables easy human-robot interaction through language. At execution time, the user can influence the behavior of the robot by simply talking to it. Our main contributions can be summarized as follows:

- We propose a Multimodal Policy Network (MPN), an approach that fundamentally combines language, vision, and motion control in to a single differentiable neural network that can learn the cross-modal relationships found in the data.
- We empirically show that our model is capable of generating task-specific robot controllers given demonstrations of a task containing natural language and visual descriptors

In order to outline our problem statement, we contrast our approach to Imitation learning (Schaal, 1999) which considers the problem of learning a policy $\pi$ from a given set of demonstrations $\mathcal{D} = \{\mathbf{d}^0, .., \mathbf{d}^m\}$. Each demonstration spans a time horizon $T$ and contains information about the robot states and actions, e.g., demonstrated sensor values and control inputs at each time step. Robot states at each time step within a demonstration are denoted by $\mathbf{x}_t$. In contrast to other imitation learning approaches, we assume that we have access to the raw camera images of the robot $\boldsymbol{I}_t$ at teach time step, as well as access to a verbal description of the task in natural language. This description may provide critical information about the context, goals or objects involved in the task and is denoted as $\mathbf{s}$. Given this information, our overall objective is to learn a policy $\pi$ which imitates the demonstrated behavior, while also capturing semantics and important visual features. After training, we can provide the policy $\pi(\mathbf{s}, \boldsymbol{I})$ with a different, new state of the robot and a new verbal description (instruction) as parameters. The policy will then generate the control signals needed to perform which take the new visual input and semantic context into account.

## 2 BACKGROUND

A fundamental challenge in imitation learning is the extraction of policies that do not only cover the trained scenarios, but also generalize to a wide range of other situations. A large body of literature has addressed the problem of learning robot motor skills by imitation (Argall et al., 2009). The majority of these approaches focus on learning functional (Ijspeert et al., 2013) or probabilistic representations (Maeda et al., 2014) of motion trajectories. Once such a model is learned, an input state vector is used to adapt the original motion to changes in position, orientation, or force. However, the state vector has to be carefully designed in order to ensure that all necessary information for adaptation is available. Neural approaches to imitation learning Pomerleau (1989) circumvent this problem by learning feature representations that are best suited for the task. Extracting feature information from rich data sources such as natural language and visual data for motion control has an extensive history. The work presented in (Arumugam et al., 2019; Burke et al., 2019; Hristov et al., 2019; Misra et al., 2018) focuses on sequencing manipulation tasks or choosing when to switch skill based on language and/or vision input from the environment. However, these approaches assume that underlying motion primitives are available that actuate the robot in the form of a motion planner or goal-directed controller. Fine grained robot control has been learned from high-level task descriptions in recent work presented by Chang et al. which utilizes robot trajectories from demonstrations by learning a parameterized neural policy from visual perception of the environment. While not using natural language to specify the target, this work outlines the importance of combining robot motions with other modalities. The work presented in Sung et al. (2015) combines natural language, point-cloud perceptions of the environment and trajectories into a joint embedding that locates tasks and trajectory representations in close proximity to each other in the latent space. At inference time, a control trajectory is generated by locating the task in the latent space and selecting an appropriate control policy.

Modern variants of this line of research leverage recent progress in training convolutional neural networks in order to train increasingly complex policies from raw (visual) sensor inputs. Building upon the same basic framework, the work in Finn et al. (2017) investigates how meta-learning can be used to learn rapidly adaptable policies. More specifically, meta-learning aims at learning policy parameters that can quickly be fine-tuned to new tasks. While very successful in dealing with visual and spatial information, these approaches do not incorporate any semantic or linguistic component into the learning process. Creating policies that can be conditioned on natural language is one potential pathway to overcome this limitation. Several works have investigated the idea of combining natural language and imitation learning: Nicolescu & Mataric (2003); Gemignani et al. (2015); Cederborg

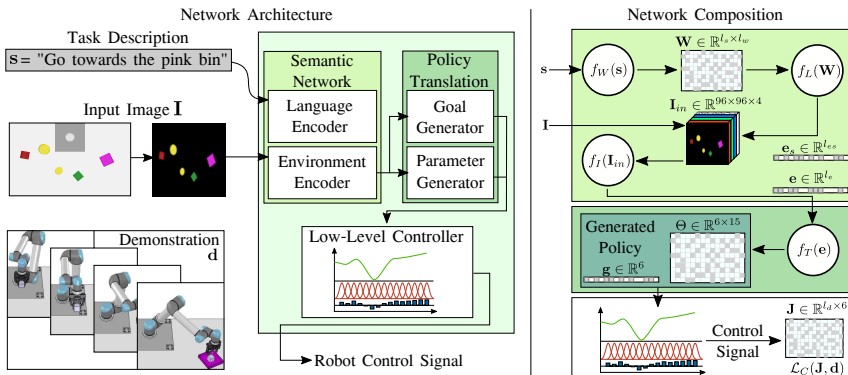

Figure 1: Description of the network architecture and compositons.

& Oudeyer (2013); Mericli et al. (2014); Sugita & Tani (2005). However, many of these approaches assume that either a sufficiently large set of motion primitives is already available or that a taxonomy of the task is available, i.e., language and motion are not trained in conjunction.

Our work is most closely related to the framework introduced in Tellex et al. (2014), which also focuses on the symbol grounding problem. More specifically, the work in Tellex et al. (2014) aims at mapping perceptual features in the external world to constituents in an expert-provided natural language instruction. Our work approaches the problem of generating dynamic robot policies by fundamentally combining language, vision, and motion control in to a single differentiable neural network that can learn the cross-modal relationships found in the data with minimal human feature engineering. Unlike previous work, our proposed model is capable of directly generating complex low-level control policies from language and vision that reassemble robot motions demonstrated during training.

## 3 MULTIMODAL POLICY GENERATION VIA IMITATION

We motivate our approach with a simple example: consider a binning task in which a robot has to drop an object into one of several differently shaped and colored bowls on a table. A human expert can teach the task to the robot providing a kinesthetic demonstration, i.e., physically maneuvering the robot through the necessary motion trajectory. However, in this example, it is critical to place the object in the correct bowl rather than only reproducing the control trajectories from the demonstrations. To this end, the human demonstrator may provide a verbal command, e.g., *"Move towards the blue bowl"* during teaching. The trajectory generation would then have to be conditioned on the *blue* bowl's position which, however, has to be extracted from visual sensing. Our approach automatically detects and extracts these relationships between vision, language, and motion modalities during learning. The result is a neural network representation that integrates all available information in order to make best usage of contextual information for better generalization and disambiguation.

Figure 1 (left) provides an overview of our method. Our goal is to train a deep neural network that can take as input a task description $s$ and and image $I$ and consequently generates robot controls. In the remainder of this paper, we will refer to our network as the MPN. Rather than immediately producing control signals, the MPN will generate the parameters for a lower-level controller. This distinction allows us to build upon well-established control schemes in robotics and optimal control. In our specific case, we use the widely used Dynamic Motor Primitives (Ijspeert et al., 2013) as a lower-level controller for control signal generation.

Given an image and a task description as input, first a so-called semantic network is utilized to combine the information from natural language with the visual perception of the robot in order to produce a joint task embedding. The joint embedding is created by converting words into a sentence embedding, which is in turn concatenated as a fourth channel to the input image. Images are provided to the network as difference images between an empty environment and the current raw camera image, resulting in an image that highlights the objects located in the environment.

This step is performed as a simple background substraction process to improve learning speed. The joint task embedding serves as a robot-independent description of the desired task. The embedding is forwarded to a sub-network, called the Policy Translation network, which synthesizes the parameters needed to fully define a low-level control policy. The resulting parameter vector can be used to execute the DMP and actuate the robot. While the MPN is activated only once per task to yield the DMP parameters, the synthesized low-level controller is continuously utilized at every time step during task execution. The following sections will introduce each part of the MPN in more detail. An in-depth overview of our architecture can be found in Figure 1.

## 3.1 SEMANTIC NETWORK

In order to extract salient information from a natural language sentence, we tokenize the sentence into a vector of words $s$. The vector $s$ is modified to have length $l_s$; sentences with fewer than $l_s$ words are zero-padded and sentences with greater than $l_s$ words are truncated. Each word is transformed into a $l_w$-dimensional word representation via the pre-trained GloVe model (Pennington et al., 2014) such that we produce a word representation matrix $W \in \mathbb{R}^{l_s \times l_w} = f_W(s)$ . We then extract the relevant $n$-grams relating to the task at hand through the use of a CNN as in Yang et al. (2015). In this method, the filters of the CNN are used to extract individual $n$-grams, such that a filter with dimension $n \times l_w$ produces a gram of size $n$. In order to determine which of these $n$-grams is relevant, we concatenate all of the convolved feature maps resulting from all filters, $m_c = [m_{c,1}, m_{c,2}, , ..., m_{c,l_s-c+1}]$, then apply max pooling such that $m'_c = \max_{0 \le i \le l_s-c+1}(m_{c,1}, m_{c,2}, ..., m_i)$. The final $n$-gram representation is built by concatenating the feature maps $s' = m'_c \forall c \in \mathcal{C}$. However, the relationship between the $n$-grams is still unknown; in contrast to prior work we leverage this information by further passing the $n$-gram map $s'$ through a two-layer fully-connected network: $e_s = \text{ReLU}(K_1\text{ReLU}(K_2 s' + b_2) + b_1)$ where $K_i$ and $b_i$ represent the kernel and bias for each of the two layers. The process of converting $W$ into $e_s$ is denoted $f_L(W)$ in Figure 1. We expand the input image $I$ with a fourth channel, composed of the sentence embedding $e_s$. To this end, we stack the sentence embedding to match the size of one input channel of the image $e'_s = [e'_s, ..., e'_s]$. The resulting image $I_{in}$ is used as an input for $f_I(I_{in})$ to generate the task embedding $e$, which is produced with three blocks of convolutional layers, composed of two convolutions, followed by a residual convolution each. The use of residual convolutions as proposed in He et al. (2015) allows the network to utilize possible accuracy gains from increased depth without increasing the complexity of the network significantly, while maintaining the property of being easily optimized. The goal of the image network $f_I()$ is to generate a joint task representation from language and environmental perception that can be further utilized to generate low-level policies.

## 3.2 POLICY TRANSLATION NETWORK

The objective of the Policy Translation network is to produce the control parameters for a low-level controller. Hence, it can be seen as a function that maps task embeddings to control parameters. Since in our case the controller is a DMP, we will first formally introduce the basics of this control framework. A DMP is fundamentally a damped spring dynamical system which produces a trajectory of joint configurations, $y \in \mathbb{R}^{d_r}$, for $d_r$ actuated robot DoFs,

$$\tau\ddot{y} = \alpha_y\left(\beta_y\left(g - y\right) - \dot{y}\right) + f\left(x; \Theta\right), \qquad \tau\dot{x} = -\alpha_x x, \tag{1}$$

attracted to the point $g \in \mathbb{R}^{d_r}$ according to the phase $x$, with constant coefficients $\alpha_y$, $\beta_y$, and $\alpha_x$ and the temporal scaling factor $\tau$. The forcing function $f$ determines the shape of the trajectory produced by the dynamical system, which we define as a linear combination of nonlinear Gaussian basis functions, $\Psi$:

$$f(x; \Theta) = \frac{\sum_{i=1}^{b} \Psi_i(x)\theta_i}{\sum_{i=1}^{b} \Psi_i(x)} x(g - y_0), \tag{2}$$

in which $\Theta \in \mathbb{R}^{d_r \times b}$ is a set of $b$ weight coefficients for $d_r$ DoFs and $y_0$ is the initial state. Most applications of DMPs for imitation learning (Schaal, 1999) directly learn a static set of weights for the forcing function from the demonstration data. At runtime these weights $\Theta$ and a goal position can be used to synthesize robot control signals. However, this assumes that a goal position has been generated by some other means, e.g., vision, kinematics, etc. In our approach, both the weight

coefficients $\mathbf{\Theta}$ as well as the goal position $\boldsymbol{g}$ are generated by the Policy Translation network. Given the task embedding $\boldsymbol{e}$, the policy translation network generates the hyper-parameters $\mathbf{\Theta} \in \mathbb{R}^{7 \times 15}$ and $\boldsymbol{g} \in \mathbb{R}^7$ for the low-level DMP. The generation of the hyper-parameters is defined as

$$\mathbf{\Theta}, \boldsymbol{g} = f_T(\boldsymbol{e}) = f_G\left(\text{ReLU}\left(\boldsymbol{W}_G \boldsymbol{e} + \boldsymbol{b}_G\right)\right), f_H\left(\text{ReLU}\left(\boldsymbol{W}_G \boldsymbol{e} + \boldsymbol{b}_G\right)\right) \tag{3}$$

where $f_G()$ and $f_H()$ are multilayer-perceptrons that generate $\boldsymbol{g}$ and $\mathbf{\Theta}$ respectively after having processed $\boldsymbol{e}$ in a single perceptron with weight $\boldsymbol{W}_G$ and bias $\boldsymbol{b}_G$. One interesting advantage of using DMPs is the fact that we can leverage a large body of research regarding their behavior and stability, while also allowing other extensions of DMPs (Amor et al., 2014; Paraschos et al., 2013; Khansari-Zadeh & Billard, 2011) to be incorporated to our framework.

### 3.3 TRAINING

The MPN, including all of its sub-networks, is trained in an end-to-end fashion and uses a single Adam (Kingma & Ba, 2015) optimizer $O_L$ for the entire network. Due to the used Rectified Linear Unit activation throughout the entire network, it does not suffer from the vanishing gradient problem and allows the use of a single, combined loss function, defined as follows:

$$\mathcal{L}_C = \lambda_c * MSE(\boldsymbol{T}, \boldsymbol{J}) + MSE(\boldsymbol{T}_{-1,:}, \boldsymbol{J}_{-1,:}) \tag{4}$$

where $MSE()$ denotes the Mean Squared Error loss function. The goal of the low-level controller is to re-create the shape of each trajectory $\mathbf{d}$ during training as well as predict the target joint configuration $\mathbf{g}$. Both of these objectives are summed in the loss function and weighted by $\lambda_c$ to maintain an equal contribution of both objectives to the overall loss. To allow the model for better generalization capabilities, we add Dropout at each stage of the network as well as a small amount of random Gaussian noise on the input image and demonstrated trajectory.

### 3.4 DETECTION OF INVALID TASKS

In its current state, the proposed model is forced to act in every possible scenario and language combination presented to the MPN. Situations that are not possible, e.g. moving towards an object that is not present in the current environment, may lead to dangerous behaviour of the robot. In order to address this problem, we extended the current model with an additional 3-layered MLP $\boldsymbol{v} = f_V(\boldsymbol{e})$ that predicts whether or not the requested task is possible. This function is based on the embedding $\boldsymbol{e}$ and performs a binary-classification regarding the validity of the task. This extension requires three additions to the previously described network structure. First, we add an adversary sentence $\boldsymbol{s}_a$ to each demonstration $\mathbf{d}$ that requests a task that is not possible given the current environment. We also introduce an additional loss $\mathcal{L}_V(\boldsymbol{v})$ that calculates the classification capabilities of the entire semantic network $f_E()$ and $f_V()$, since the ability to distinguish tasks needs to be propagated through the entire network up until this point. The optimizer utilizes sparse softmax cross entropy for exclusive single class classification. The third addition is an additional optimizer $O_E$ that optimizes the embedding network utilizing the following loss:

$$\mathcal{L}_E = \mathcal{L}_C + \lambda_v \mathcal{L}_V \tag{5}$$

We combine the controller and embedding network losses with an additional weight hyper-parameter $\lambda_v$. The addition of $\mathcal{L}_C$ to $\mathcal{L}_E$ is necessary to maintain the ability of the semantic network to generate useful embeddings for the translation network. This is also reflected in the choice of $\lambda_v$ which leaves a strong weight on $\mathcal{L}_C$. As a last step, the former optimizer is reduced to only optimize the translation network instead of the entire network, as described in the previous section.

## 4 EXPERIMENTS

In this section, we describe our experimental setup and conduct and extensive set of experiments to verify the capabilities of our proposed model. We evaluate our model in a simulated environment on a binning task in which the goal is drop a cube in one of randomly placed multiple bins. An image sequence of the task can be seen in Figure 2.

**Data Collection:** For data collection, we automatically generate random binning scenarios in which we present the robot with three to five different bowl of different color, shape and size. In total, we

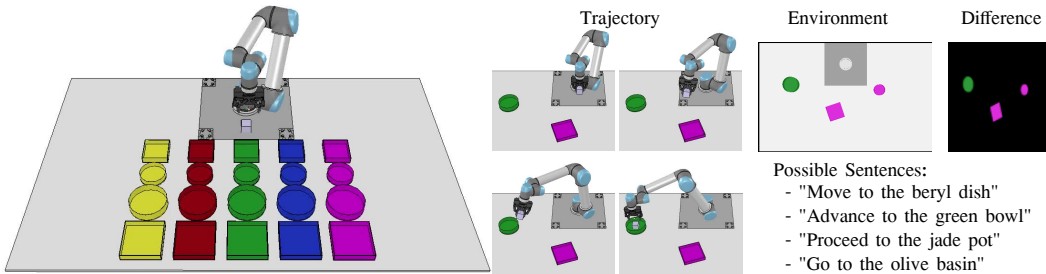

Figure 2: Overview of the experiment setup: (left) 3D environment with all possible objects with five colors, two sizes and two shapes. (middle) Example trajectory for going towards the green bowl. (right) Collected image and voice data for the given trajectory.

utilize five colors (yellow, red, green, blue and pink), two sizes (small and large) and two shapes (round and squared). This procedure provides us with 20 different objects and the ability to test different levels of ambiguity regarding the required amount of features needed to uniquely identify an object. As an example, when all four red objects are in the scene, a unique description of an object is only possible when all three features (color, size and shape) are used at the same time to describe the task. In order to generate a larger variety of different possible sentences, we conducted an IRB approved human subject study in which we presented multiple colored objects to participants and asked them to explain verbally how they are interacting with these objects in a pick-and-place task. This allowed us to extract multiple sentence templates for going towards objects, as well as multiple synonyms for actions, colors, containers and reference points. In our study, we did not restrict participants in any way, such that each individual was able to choose a description that is most natural to them. In addition, we expanded the list of synonyms by gathering additional words from publicly available synonym databases. The combination of the data from the IRB study and synonym database allows us to generate a large variety of natural task explanations for arbitrarily generated scenarios. A detailed description of the template generation can be found in Appendix A. The visual perception of the robot is provided as a top-down image from above to robot, see Figure 2. To use the images in our model, we scale them to a resolution of $96 \times 96 \times 3$ where the last three channel refer to the RGB values of each pixel. After generating a task and scenario, kinesthetic demonstrations are generated with a physics based simulation of a UR5 robot arm, taking into account inertia, weights and other properties of the robot and environment, allowing us to collect realistic movements in the simulated environment. The simulator is running with 20Hz to collect a reasonable amount of samples for each trajectory. In total, we collected over 20000 generated demonstrations as described above.

## 4.1 GENERALIZATION

Table 1: Comparison of successfully completing the binning task by using a single feature over 500 attempts.

| Required Feature | Binning Success |
|---|---|
| None (Single Object) | 96.4% |
| Color | 97.6% |
| Size | 96.0% |
| Shape | 79.0% |

Table 2: Accuracy of detecting valid actions in random environments based on ambiguity. Low: Separation based on a single feature High: Only combinations of features separate the targets.

| Ambiguity | Correctly: | |
| | Accepted | Rejected |
|---|---|---|
| Low | 99.3% | 96.6% |
| High | 70% | 66.6% |

In our binning scenario, the robot needs to stop its movement above the bowl outlined in the experiment within a radius of the bowl's center such that the dropped object from the gripper is successfully placed inside the bowl. We utilized two different bowl sizes in these experiments, large and small, with a 17.5 cm and 12.5 cm diameter respectively. The object that is to be delivered, a cube, has an edge length of 5 cm. All experiments were conducted by generating new random scenarios with new environments, images, and sentences corresponding to the generated task.

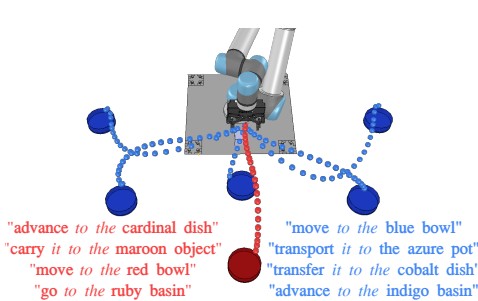
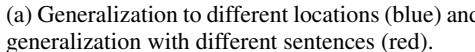

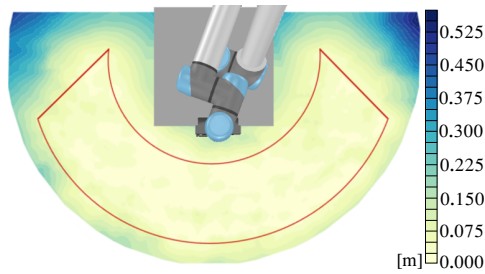

(a) Generalization to different locations (blue) and generalization with different sentences (red).

(b) Predicted goal error in meters depending on the target position. The area highlighted in red shows the area from which training data were sampled.

Figure 3: Generalization capabilities (Figure 3a) and positioning error (Figure 3b).

In our first experiment, we evaluated the success rate of the object delivery by using only a single necessary feature. The results of this set of experiments can be seen in Table 1. Each test was conducted on an equal amount of small and large bowl with 250 attempts each, resulting in 500 attempts for each shown feature. Except from the first row, all features were tested in scenarios with 3 bowls, one being the target and two serving as distraction. Based on the reported success rate, the robot is capable of successfully achieving a task with at least 96% probability except from when the only distinguishing feature is the shape (round or square), in which the success rate drops to 79%. This drop in successful task completion in these cases is due to the chosen image resolution of the environment in which distinguishing small round from small squared bowl is a particularly challenging task that was chosen on purpose to test the limits of our approach.

The generated parameters of the low-level DMP controller – the weights and goal position – must be sufficiently accurate in order to successfully deliver the object to the specified bin. A set of weights for the first four dimensions of a DMP controller can be seen in Figure 4b. The figure shows the generated weights for the movement to two different objects, one of which is closer to the robot than the other as well as being on different sides of the robots. We quantify the accuracy of the parameter generation by computing the Euclidean distance between the ground truth target location and the end effector position of the robot, based on the predicted joint configuration. For this, we generated 6000 positions on a grid that were equally distributed inside the physically reachable work space of the UR5 robot. The comparison between the end effector position, calculated with forward kinematics, and the target position can be seen in figure 3b. Within the area used to generate training data, the robot predicts the correct position with well under 5cm error, which is precise enough for the tested binning task. Additionally, the model is able to accurately generalize to target positions located outside of the training area, with the error increasing as the distance from the training area increases. The proposed addition of classifying if a requested task is possible was evaluated on random environments with low and high ambiguity regarding the number of separating features. In an environment with low ambiguity, a single feature is enough to tell targets apart, where as in environments with high ambiguity, multiple features are needed for each object to tell them apart. The results of this test are shown in Table 2. In addition to generalizing to different bowl locations, the model is also capable of generalizing to changes in the verbal task description. This is important when interacting with different users that may describe the same task with different words. In Figure 3a we show the spatial and verbal generalization capability of our model. Since color is a key component of our verbal task descriptions, we expect that the robot is able to generalize to different color shades, something that can be caused by variations in illumination. In this experiment, we changed the colors of our green objects to different shades of green. Additionally, we also change the color components towards other bowls by increasing the red and/or green component. We empirically evaluated the MPNs ability to incorporate these changes in the selection of a target object. An example of the changes we made can be seen in Figure 5. In that scenario the robot chooses the dark green object over an object that added a blue component. However, when tasked with going to the *large* green object, it also moves towards the larger object with increase blue component. This experiment shows that our network can combine information from multiple modalities to disam-

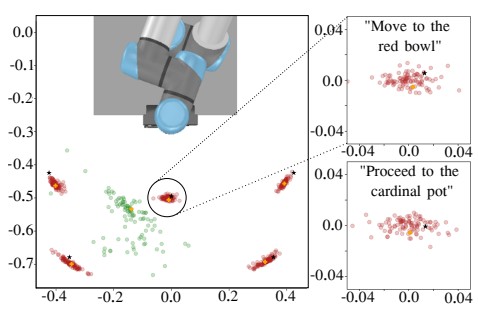

(a) Stochastic Forward Passes: Certainty of the predicted goal position for an existing object at different locations (red) and an object that is not existing (referred to as green, but is central red).

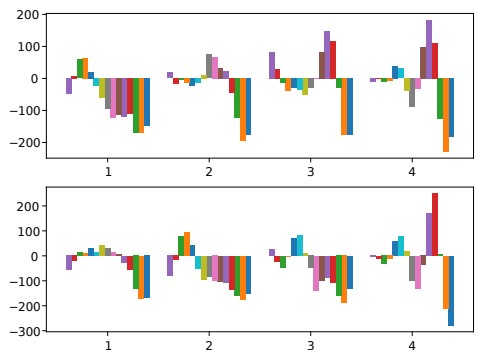

(b) Generated DMP weights for two different controllers approaching objects located on opposite sides of the robot

Figure 4: Effects of language and environment on generated controllers and target prediction.

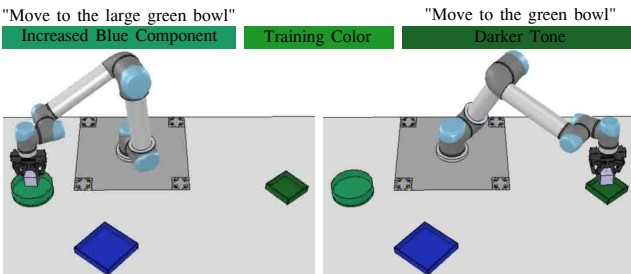

Figure 5: Sensitivity to different shades of green. During training, colors were fixed as well as the illumination. Here we show the robot's ability to account for small color changes.

biguate a situation, i.e., the greener object is chosen when no size is defined, while the slightly bluer, larger object is chosen when a size is part of the verbal description.

## 4.2 UNCERTAINTY

We leverage recent theoretical insights in order to generate uncertainties via probabilistic outputs from our trained MPN. In particular, it was shown in Gal & Ghahramani (2015) that neural network learning using the Dropout method is equivalent to a Bayesian approximation of a Gaussian Process modeling the training data. In each of the forward passes, we randomly drop neurons from the network as done in the Dropout algorithm. However, in this case, the neurons are dropped at inference time and not at training time. The generated samples form a possibly complex distribution represented as a set of outputs of the neural network. By analyzing this set we can glean important information about the uncertainty in our networks outputs. Figure 4a shows the application of stochastic forward passes on the predicted goal position in Cartesian space generated by using a forward kinematics on the predicted goal configuration of the robot. As can be seen in the picture, the network is certain about the position of five red objects on the table in five different tasks. The variance of all forward passes is below 5cm, which allows for a successful binning task. However, when only providing a red object in the environment and asking the robot to move to a green object, the uncertainty drastically increases. This can be seen in the green scatter plot, which shows positions of the green bowl over 100 stochastic passes. Based on the distribution of the predicted goals, it becomes apparent that the green object is not available in the current environment. These types of analyses allow for granular decisions regarding task execution success to be made.

## 4.3 DYNAMIC ENVIRONMENTS

It is desirable for robots to be able to cope with dynamically changing environments, particularly when a human is in the loop. In this experiment, we evaluate the robot's ability to adapt its generated

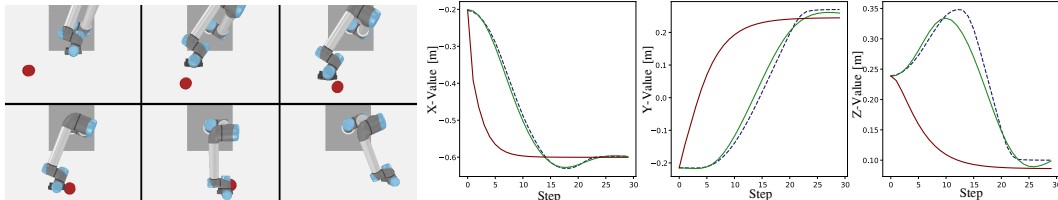

(a) Dynamic Environment: Changing bowl location while reaching target

(b) Following a learned trajectory: Demonstrated (Blue), DMP control (Green), Proportional control (Red)

Figure 6: Evaluation of our approach for dynamically changing environments (a) and the ability to adhere to demonstrated trajectory shapes during interaction (b)

Table 3: Average success rate for placing the object in 250 randomly generated environments.

| Objects | Required Features | Dataset Size | | | | Syn | No Syn |
| | | 20000 | 10000 | 5000 | 1000 | 20000 | 20000 |
|---|---|---|---|---|---|---|---|
| 1 | Small | **92.8**% | **93.6**% | 45.0% | 54.8% | **92.8**% | **100.0**% |
| 1 | Large | **100.0**% | **96.8**% | 61.6% | 69.2% | 91.2% | **100.0**% |
| 3 | Color, Small | **96.4**% | **91.6**% | 33.2% | 20.8% | 22.0% | **99.6**% |
| 3 | Color, Large | **98.8**% | **98.0**% | 50.8% | 29.2% | 26.0% | **100.0**% |
| 3 | Size, Small | **92.8**% | 40.0% | 6.8% | 4.0% | 77.2% | **99.6**% |
| 3 | Size, Large | **99.2**% | 37.6% | 23.6% | 21.2% | 15.6% | **100.0**% |
| 3 | Shape, Small | 70.0% | 8.0% | 14.4% | 6.8% | 56.0% | 82.4% |
| 3 | Shape, Large | 88.0% | 20.4% | 24.4% | 9.6% | 64.0% | 72.0% |

policy to a dynamically changing environment by asking the robot to drop an object in a constantly moving bowl. During data collection and training, the robot was only provided with examples from static environments, such that it was enough to generate a DMP once at the beginning of each interaction. However, to adapt to a changing environment, a new DMP needs to be generated for each time step. Figure 6a shows such a scenario, in which the red bowl is moving on an arc from left to right around the robot by moving 1.5 cm in each step. For this experiment, we utilize the same model as for previous experiments without having trained it for dynamically changing environments. At each time step, the same sentence $s$ is combined with the new environment image $\mathbf{I}$, generating a new policy by providing the parameters for an updated DMP. As can be seen from the image sequence in Figure 6a (a), the robot is successfully able to adapt to the changed bowl position.

### 4.4 TRAJECTORY RECONSTRUCTION

In our work, we chose a DMP as a low-level controller for the MPN model to give the robot the ability to not just learn to approach a predicted goal position, but to also reassemble the shape of demonstrated trajectories. This ability is essential in scenarios in which the trajectory shape encodes additional information, e.g. object avoidance or a certain way in which an object needs to be approached. Figure 6b (b) shows the MPN's ability to generate trajectories that are similar to what was shown during training. The dashed blue line shows the position of the tool center point (in x, y, z coordinates) of a demonstrated trajectory from the test set, the green line shows the respectively generated trajectory, executed by the DMP controller and the red line demonstrates the executed trajectory when using a proportional controller. The movement along the Z-axis of the trajectory clearly shows a different behaviour of the robot when using a proportional controller. On average, the difference between the tool center point position and the demonstrated trajectory is 1.6cm and 19.1cm when using the DMP and proportional controller, respectively.

### 4.5 ABLATION STUDY

**Dataset Size**  Even though we are able to generate a large amount of artificial demonstrations in simulation, the ability of the MPN to train on fewer data is desirable. For this purpose we looked at the performance of the MPN when being trained with less than 20000 demonstrations, see Table

Table 4: Ablation over network structure. Average success rate for placing 250 objects using n-grams of size 2, 3, 4 and 5. The last demonstrates the influence of using residual layers in the image processing pipeline.

| Objects | Required Features | Original | Network Changes | | | | |
|---|---|---|---|---|---|---|---|
| | | | NG 2 | NG 3 | NG 4 | NG 5 | NRes |
| 1 | Small | **92.8**% | 31.6% | 30.0% | 32.8% | 15.2% | 56.6% |
| 1 | Large | **100.0**% | 52.4% | 40.4% | 56.8% | 22.8% | 74.4% |
| 3 | Color, Small | **96.4**% | 32.4% | 21.2% | 29.2% | 18.0% | 59.2% |
| 3 | Color, Large | **98.8**% | 56.8% | 43.2% | 59.2% | 26.8% | 74.4% |
| 3 | Size, Small | **92.8**% | 31.2% | 26.0% | 4.4% | 14.0% | 56.8% |
| 3 | Size, Large | **99.2**% | 53.2% | 44.0% | 19.2% | 26.0% | 71.6% |
| 3 | Shape, Small | **70.0**% | 4.8% | 8.4% | 5.2% | 14.4% | 45.2% |
| 3 | Shape, Large | **88.0**% | 22.4% | 42.0% | 10.8% | 24.4% | 68.4% |

3. As when testing the spatial generalization capability of the network in section 4.1, we conducted experiments with various combinations of features, separated by their success rate with regards to the object size. The trained MPN seems to be working better with larger objects, which again, might be related to the chosen image size. However, the experiments showed that when identifying the location of a single object or a scenario in which the color is sufficient to distinguish targets, it does not significantly benefit from more than 10000 training data. In addition to the amount of training data, we also trained a model without augmenting the the training data with synonyms, which is shown in the last two columns. This model was tested on data using synonyms (2nd last column) and data not using synonyms (last column). As expected the problem becomes easier when no synonyms are used. However, this model shows the ability of the Glove word embeddings to embed words with a similar meaning closer to each other, resulting in a partially usable model.

**Network Structure**    In addition to the size and variety of the dataset, the structure of the network is an important component of our approach. In this section, we compare different choices with regards to the network structure. Table 4 analyzes the performance of different n-gram sizes as compared to the original model. As expected, using a single n-gram size performs significantly worse across all tested sizes as compared to our original model using n-gram sizes of 1, 2, 3 and 5 concurrently. However, the results suggest that smaller n-gram sizes are better at capturing cases in which a single feature is enough to uniquely describe an object where as large n-grams seem to loose the ability to focus on the important part of the sentence.

In addition to process language, our approach is able to ground sentences with the current environment perception. ResNet is a common model for tasks related to computer vision (He et al., 2015) and achieved its performance by introducing residual layers in the CNN structure. In the right-most column of table 4 we analyzed the influence of the residual sections of our network by replacing them with max-pooling layers to maintain a similar output structure. As can be seen form the results, the residual units have a significant influence on the overall performance of the network.

## 5 CONCLUSION

In this work, we presented an imitation learning approach combining language, vision, and motion. A neural network architecture called Multimodal Policy Network was introduced which is able to learn the cross-modal relationships in the training data and achieve high generalization and disambiguation performance as a result. Our experiments showed that the model is able to generalize towards different locations and sentences while maintaining a high success rate of delivering an object to a desired bowl. In addition, we discussed two extensions of the method that allow us to obtain uncertainty information from the model by either learning a separate classifier or utilizing stochastic network outputs to get a distribution over the belief.

Finally, we hope to further expand the verbal fidelity of our model by adding the ability to utilize relational object descriptions to parameterize the task. Using the full range of natural language descriptions will give us the ability to ground additional constraints into robot control.

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

## A  Voice Template Generation

To train our neural network, we require a large amount of training data $\mathcal{D}$ of sentences $s$, images $\mathbf{I}$ and respective sensor readings $x_t \forall t \in T$ from the respective robot movement. While generating random scenarios and actuating the robot in a simulator can easily be done, generating sentences for random scenarios is a more challenging task. Not only do sentences need to describe a target object uniquely given all other objects in the scene, but they also need to reflect grammatically correct and

Table 5: Words gathered from the human-subject study witch their common synonyms.

| Base Word | Used Synonyms | | | | |
|---|---|---|---|---|---|
| round | round | circular | | | |
| square | square | rectangular | | | |
| small | small | tiny | smallest | petite | meager |
| large | large | largest | big | biggest | giant | grand |
| red | red | ruby | cardinal | crimson | maroon | carmine |
| green | green | aquamarine | olive | jade | chartreuse | beryl |
| blue | blue | azure | cobalt | indigo | turquoise |
| yellow | yellow | amber | bisque | blond | gold | sand |
| pink | pink | salmon | coral | rose | blush |
| cube | cube | object | piece | dice | die |
| place | place | put down | deposit | lay down | set down |
| | plant | release | | | |
| goto | go to | move to | advance to | progress to | carry to |
| | transport to | transfer to | | | |
| bowl | bowl | basin | dish | pot | |
| left | left | port | | | |
| right | right | starboard | | | |
| <target> | <color> <size> <shape> <bowl> | | | | |
| <source> | <cube> | | | | |

realistic sentences. To achieve the generation of sentences for random scenarios, we conducted a human-subject study in which we presented multiple differently colored objects to the participant with the task to assemble them together while describing their actions. To prevent any bias towards certain explanations, we only showed the participants the initial state as well as the final result of the assembly task without further explanations. Participants were given a few tries to assemble the objects correctly before starting the experiments. The goal of the study was to collect sentences from different participants that describe common tasks like reaching, pushing, grasping, placing and insertion for a variety of differently shaped and colored objects. As a result, we received two general sentence templates for the reaching task as well as set of synonyms for colors, shapes and objects. The two basic templates are as follows: (1) "<place> the <source> in <target>" and (2) "<goto> to <target>". The placeholders as well as the respective words used to replace them are shown in Table 5.

**Sentence generation** Each random scenario contains between three and five objects. One of these objects is chosen at random as the target for the reaching task. Before being able to generate a sentence, the unique properties of the object need to be extracted. Unique properties can be related to their global position, color, size, shape or combinations of these features. Using color, size and shape to describe an object will always result in a unique object description. However, results from the human-subject study showed that participants used the smallest set of features necessary to uniquely describe and object. Given the type and location of all objects in the scene, we identify the smallest set of features necessary to describe the object. In case multiple sets qualify as the smallest set of features, a random selection is made. The selection of features as well as an object description is used as the <target> placeholder while replacing features as needed. Taking the different feature sets into account, we are able to generate 180,000 different sentences of which we are using 20,000 for training. Depending on the randomly generated environment, the same sentence may be used multiple times for different scenarios.

