# OpenReview forum: "Imitation Learning of Robot Policies using Language, Vision and Motion"
_ICLR.cc/2020/Conference — Reject_

### Official Review · AnonReviewer1 · 2019-10-23
**Official Blind Review #1**

**Rating:** 6

**Review:**

This work uses imitations learning (from synthetic data) to train a deep model which takes a natural language instruction, and a visual representation of a robot's environment, and outputs a trajectory for the robot to follow which executes this instruction.  The work focuses on a robotic pick-and-place task, where the instruction indicates which of the available bins an item should be placed in.  In addition to the trajectory model, a second model is trained which allows the agent to predict whether a given command is actually feasible (i.e. whether the target bin exists).  Empirical results show a reasonably high success rate in placing objects in the bin specified by the instruction, though there is still room for improvement in cases where the shape o a combination of features is important to the selection of the correct bin.

Rather than mapping directly from instructions and observations to control signals, the model trained in this work translates from an instruction, and an image of the agent's environment, to the parameters of a DMP controller.  The network therefore outputs the entire motion for the task in a single inference pass.  This approach would have advantages and disadvantages.  The DMP formulation ensures that the resulting trajectory is relatively smooth.  It also means that the network outputs a distinct goal configuration, which the DMP should reach (assuming the goal is feasible) regardless of the other motion parameters.  The use of a DMP output space, however, limits the model to generating relatively simple, goal-directed motions, and does not allow the agent to adapt to changes in the layout of the environment (which would only be observed in the static visual input).

As other work has considered visual instruction following (e.g. Misra et. al. "Mapping Instructions and Visual Observations to Actions with Reinforcement Learning") it would strengthen this work considerably to see a direct comparison between this method and existing approaches.  It is likely that the approach presented in this work is better suited to the specific problem of robot control, but it would be helpful to see if learning a low-level control policy directly can be successful in this context.

The work needs to expand on the discussion in the second paragraph of section 4, where human annotators were used to generate natural language instructions for different tasks.  The paper suggests that this data was not used directly to train the model, but was instead used to build a template for generating natural language instructions.  What this template looks like, and how it was constructed based on the human-generated data, remains unclear, and needs to be described in much more detail.

**Experience Assessment:**

I have published one or two papers in this area.

**Review Assessment: Checking Correctness Of Derivations And Theory:**

I assessed the sensibility of the derivations and theory.

**Review Assessment: Checking Correctness Of Experiments:**

I carefully checked the experiments.

**Review Assessment: Thoroughness In Paper Reading:**

I read the paper thoroughly.

---

> ### Author Response · Authors · 2019-11-15
> **Reviewer #1 Response**
>
> Summary:
> Thank you for your review. We added additional experiments in section 4.3 and 4.4 to outline our methods ability to react to dynamically changing environments and reassemble the shape of demonstrated trajectories. We also added an appendix that goes into further detail about the human-subject study and how we generate sentences for our random training scenarios. In contrast to your suggested reference, we present an approach that generates low-level robot controllers based on language and images instead of generating the next action from a discrete set of action.
>
> Detailed Response:
> Thank you for your constructive review. As pointed out in your review, we use the network to output the parameters for a DMP describing the entire motion and utilized it to actuate the robot. However, our approach is capable of generating a new DMP at each time step to adapt to potential changes in the environment. To outline this ability further we added an additional experiment in which the robot is attempting to approach a moving object. In each time step, we moved the object by 1.5cm along a predefined trajectory and regenerated the DMP to obtain the updated trajectory. Without needing additional training, the experiments show that our approach is able to adapt to dynamic environments. Please refer to section 4.3 for further details.
>
> While our neural network utilizes a DMP, which is ultimately converging to a goal position (assuming the goal is feasible), the forcing term allows the DMP to actuate the robot such that it adheres the shape of the trajectory learned from the demonstrations instead of just approaching the goal position. In order to demonstrate this behaviour, we added an additional figure (Figure 6 (b)) to outline the difference between using a DMP as our low-level controller and a goal-directed controller. Please refer to section 4.4 for further details.
>
> To generate the amount of training data necessary to successfully train our network, we could not exclusively rely on human demonstrations. For this reason, we conducted a human-subject study to collect sentences and words related to pick-and-place tasks and further utilized this information to create a sentence generator. In addition to the collected data, we expanded the list of words with common synonyms from respected NLP databases. As requested, we added appendix A describing the basic templates as well as all synonyms together with the sentence generator in greater detail.
>
> The reference suggested in your review has been incorporated in our paper. The paper presents an approach in combining language and images for robot control by generating multi step action sequences. Similar to our method, the work proposed in the reference uses unrestricted natural language to describe the task and combines it with a visual perception of the environment. In contrast to our work, the reference generates an action from a discrete set of actions instead of generating a continuous low-level controller that directly actuates a robot. Since our work focuses on generating low-level control policies from latent task representations that can directly be used to actuate a robot with respect to demonstrated joint trajectories for a single task, we feel that a comparison to other papers in this area would be miss-leading to readers since a core contribution of our work is to translate high-level semantics from language and vision into low-level control policies rather than learning to generate actions for a subsequent task planner. While the suggested work is interesting in regards to its ability to generate multi step plans, we do not currently look at performing tasks that require more than one action.

---

> > ### Comment · AnonReviewer1 · 2019-11-15
> > **Dynamic Environments**
> >
> > Thank you for taking the time to address these comments.
> >
> > The closed loop control experiments are very helpful.  I think it would be worth conducting a similar experiment in environments where the trajectory itself (not just the goal) needs to change over time, to avoid dynamics obstacles.  This would show that the policy is able to appropriately adjust the shape parameters of the DMP in response to changes in the environment.

---

### Official Review · AnonReviewer3 · 2019-10-23
**Official Blind Review #3**

**Rating:** 3

**Review:**

*Summary

The paper describes a new end-to-end imitation learning method combining language, vision, and motion.
A neural network architecture called Multimodal Policy Network is proposed. That can extract internal representations from language and vision to condition the generated motions.
It enables an end-user to influence a robot's policy through verbal communication.
The experiments demonstrate the generalization performance of the method. That can generate behaviors towards different goals depending on different sentences.

*Decision and supporting arguments

I think the paper is just below the borderline. The reason is as follows.

The concern is about evaluation. They demonstrated the method could work, and the robot can move to appropriate goals. However, there is no comparative methods in the experiment.
Related to this point, the problem was not identified in the Introduction.
The authors might assume that introducing language into behavioral cloning itself is qualitatively new work. However, such a study has a long history.
For example, please refer to Tani's pioneering works.
Sugita, Yuuya, and Jun Tani. "Learning semantic combinatoriality from the interaction between linguistic and behavioral processes." Adaptive behavior 13.1 (2005): 33-52.

The author should specify a current challenge or problem in pre-existing studies about imitation learning with language input, clarify their claim, and give empirical support for the claim.


**Experience Assessment:**

I have read many papers in this area.

**Review Assessment: Checking Correctness Of Derivations And Theory:**

N/A

**Review Assessment: Checking Correctness Of Experiments:**

I assessed the sensibility of the experiments.

**Review Assessment: Thoroughness In Paper Reading:**

I read the paper at least twice and used my best judgement in assessing the paper.

---

> ### Author Response · Authors · 2019-11-15
> **Reviewer #3 Response**
>
> Summary:
> Thank you for your review. We outlined the problem and novelty of our work more carefully in the introduction and background section. The novelty of our work lies in proposing an approach that fundamentally combines language, vision and motion in an end-to-end fashion, thereby grounding language in motion with minimal human feature engineering.
>
> Detailed Response:
> We would like to thank the reviewer for outlining the excellent reference to Tani's pioneering work, which has been included in the paper. We incorporated additional references and altered our introduction and background section to better outline the problem statement and its relevance in light of current advancements (section 1 and 2).
>
> The main contribution of our work is to look at language, vision and control as a fundamentally connected problem. This allows us to not only ground language in the environment like [1][2] but to ground language in control policies together with the additional information gained from the vision component. We agree that including language into behavioural cloning is fundamentally not a new idea, as outlined by your reference to [3] and other recent work in the same area [4]. However, we propose a system that fundamentally combines these three lines of research that goes beyond previous work by introducing a fully differentiable approach that grounds language and vision in robot motion learned from demonstrations with minimal human feature engineering. Another contribution of our model is that we are able to generate the parameters for a continuous controller while many other approaches use discretization to either limit their input space [3] or output space [1][5]. Our method builds upon a rich literature on learning embeddings for vision [6], language [4] and tasks [7], allowing us to connect these lines of research to create an end-to-end approach that generates a task-specific continuous controller capable to seamlessly adapt to different tasks.
>
> Our experiments justify the feasibility of our approach by demonstrating the ability of our MPN model to generalize towards different sentences and environments (Section 4.1) while generating trajectories similar to the demonstrated behaviours (Section 4.4). Preliminary results on dynamically changing environments suggest the methods ability to adapt to changes in the environment (Section 4.3) which allows the model to work in collaboration with human partners. Especially when humans are working in close proximity with robots, it is important that robots can adapt to changing conditions while also assessing the feasibility of the given tasks (Section 4.2).
>
> References:
> (1) "Disentangled Relational Representations for Explaining and Learning from Demonstration" Hristov et al
> (2) "CLEVR: A Diagnostic Dataset for Compositional Language and Elementary Visual Reasoning" Johnson et al
> (3) "Learning semantic combinatoriality from the interaction between linguistic and behavioral processes." Sugita et al
> (4) "Grounding natural language instructions to semantic goal representations for abstraction and generalization" Arumugam et al
> (5) "Mapping Instructions and Visual Observations to Actions with Reinforcement Learning" Misra et al
> (6) "Learning Deep Parameterized Skills from Demonstration for Re-targetable Visuomotor Control" Chang et al
> (7) "Deep Multimodal Embedding: Manipulating Novel Objects with Point-clouds, Language and Trajectories" Sung et al

---

### Official Review · AnonReviewer2 · 2019-10-23
**Official Blind Review #2**

**Rating:** 1

**Review:**

The paper addresses the problem of using multiple modalities for learning from demonstration. Approaches that take in task or joint space data to learn a policy for replicating that task are numerous. Doing the same with multiple modalities involved, in particular vision, language and motion, has only been recently considered, so this is a timely paper.

The core contribution is pretty well summarised by the architecture in figure 1, which involves a combination of encodings of the words and sentences, images and parameters of a DMP in order to generate movement commands from a high level instruction.

Unless I have missed something in the experimental setup, all of the considered task variations are movement commands of the form <Move> to <Object>. The network setup allows for synonyms of two kinds, so <Move> can be replaced by numerous verbal synonyms such as advance and go, and the object can be specified in terms of shapes, colors and so on, but otherwise this is the only specification of the task. This has been addressed in the recent literature using neural network architectures similar to the one being proposed here, e.g., see the following papers. These papers already solve the proposed problem and provide similar explanations. It would be helpful to see comparative discussion with respect to those methods and a clear statement of novelty with respect to such prior work:
[R1] M. Burke, S. Penkov, S. Ramamoorthy, From explanation to synthesis: Compositional program induction for learning from demonstration, Robotics: Science and Systems (R:SS), 2019.
[R2] Y. Hristov, D. Angelov, A.Lascarides, M. Burke, S. Ramamoorthy, Disentangled Relational Representations for Explaining and Learning from Demonstration, Conference on Robot Learning (CoRL), 2019.

An interesting feature in R2 that the authors do not explicitly address here is the issue of relational specifications in the language, e.g., in addition to saying "move to the red bowl", we may also wish to say "place on top of red block". In the way that MPN is currently set up to map from the language input directly to hyperparameters of the DMP, and considering the embedding structure, it is not clear if MPN is capable of handling such specifications. If so, the claim of generalisation on the language input should be stated more clearly.

The ablation study is setup somewhat differently than what I would have expected. The authors consider the effect of changing the training set size and if the language input includes synonyms or not. Those two aspects seem to produce the expected results. It would also be interesting to see an ablation study in the sense of replacing or removing aspects of the architecture to see its relative effect on the overall model performance. So, for instance, if one did not have a DMP with the hyperparameters being estimated by a network and instead had a more straightforward encoding of where to move to - does it make a difference and how much? Likewise, how much performance benefit, if any, is being derived from an uninterpreted image I being combined as described in the embedding as opposed to an alternative that detects an objects and combines that position differently. The paper would have been stronger if such architectural choices were better justified and also demonstrated in the experiments.




**Experience Assessment:**

I have published in this field for several years.

**Review Assessment: Checking Correctness Of Derivations And Theory:**

N/A

**Review Assessment: Checking Correctness Of Experiments:**

I carefully checked the experiments.

**Review Assessment: Thoroughness In Paper Reading:**

I read the paper thoroughly.

---

> ### Author Response · Authors · 2019-11-15
> **Reviewer #2 Response (Part 1)**
>
> Summary:
> Thank you for your review. We added an appendix that outlines the generation of sentences for random training scenarios in further detail. Both of the references mentioned in your review propose a method to extract action sequences from demonstrations that is used in subsequent low-level controllers. In contrast, our work focuses on dynamically generating low-level controllers in an end-to-end fashion for different tasks. While action sequences are a future goal of our research, we currently do not address this problem in our work.
>
> Detailed Response:
> Thank you for your in-depth review. As pointed out by your review as well as due to concerns from other reviewers regarding our sentence generation, we added appendix A explaining the collection, structure and generation of sentences in further detail. In total, we are able to generate ~180,000 unique sentences which are utilized depending on the environment of the robot. Please refer to the appendix for further details on how sentences are generated.
>
> As compared to R1, our work utilizes natural language to convey instructions to the robot, allowing users to use a natural and intuitive interface for interaction as compared to pre-programming a sequence of desired actions. Furthermore, our approach generates low-level control policies in form of a DMP that allows the robot to resemble the shapes of demonstrated trajectories instead of just going to the predicted goal. To address this feature further, we added a short experiment in section 4.4 comparing the use of a DMP over a goal-directed controller, showing the ability of the DMP to recreate demonstrated trajectories. The work proposed in R1 is focusing on generating action sequences and switching between a set of goal directed motion controllers. In our work, we focus on automatically generating a unique controller for each task at hand from natural language conditioning that generates motions similar to what was demonstrated, even in dynamically changing environments. The work in R1 presents a methodology to run different controllers sequentially to achieve a multi-staged inspection task from pre-defined action sequences with great success. As of right now, we are not focusing on addressing multi-staged tasks, but it is certainly a future direction of our research.
>
> The work presented in R2 mainly focuses on learning to ground inter-object relationships from visual environment perceptions to their respective words. While the work includes a robot experiment, robot control is done by predicting a goal position and using a proportional controller to reach the goal position while disregarding any information from possibly demonstrated movements. While a more versatile language model is a future objective of our work, we are currently not able to use relational descriptions in our work except from global descriptors like "left" or "right-most". We will incorporate the results from R2 in future work to further enhance our method. However, our main contributions is to directly translate unrestricted natural language into low-level control policies for robot actuation. While language grounding is an essential part of our work, we focus on translating high-level semantic task descriptions into complex low-level controllers that reassemble the demonstrated trajectories of the task. Our proposed method has the benefit that everything from the high-level semantics to the low-level control is a single differentiable model that can be trained end-to-end.

---

> > ### Author Response · Authors · 2019-11-15
> > **Reviewer #2 Response (Part 2)**
> >
> > Our network structure was influenced by an automatic parameter search process. However, we agree that a better ablation study regarding the network structure benefits the justification of our model. We extended the existing ablation study with an additional section with regards to certain design choices within the network. We compared the n-gram sizes as well as the use of residual layers in the image processing part of the neural network in section 4.5. Additionally we want to underline the use of a DMP as compared to a simple proportional controller allowing us the teach the robot to reassemble entire trajectory shapes as learned from the initial demonstrations. We further added an additional figure (Figure 6 (b)) to outline this feature of our approach (see section 4.4). While we did not make explicit use of this ability in our current work, we intend to show how it can be used to perform object avoidance or train the robot to approach objects from different sides in future work. Both of these cases require the robot to be actuated differently while having the same goal position, which could not be solved by proportional controllers, thus the choice of a DMP. The results presented in section 4.4 show promising results for our model to further utilize the information extracted from the demonstrated trajectory shapes in future experiments.
> >
> > Combining a difference image with the language in the network as described allows the network to independently learn the features necessary to ground natural language in the perceived environment. Using a manual approach for feature extraction would certainly be possible, but it would probably limit the network's ability to ground the language in the environment while requiring a considerable amount of time to manually engineer the necessary features. To create a comparative method, features would need to be extracted from language as well as the image in a manual fashion before being manually combined to form a informative representation that could be used to be translated into a respective low-level controller. While it is not impossible to use such an approach, we are currently unable to provide information on how the performance would differ as compared to our proposed MPN due to the extensive amount of work required to create a manual feature extractor as described above.

---

### Author Response · Authors · 2019-11-15
**Thank you for the reviews: General Response**

We would like to thank all reviewers for their constructive and helpful feedback on our paper. An updated version of the paper is uploaded. For individual responses to the reviews, please see our respective posts. However, our changes to the paper can be summarized as follows:

- We re-formulated parts of the introduction (Section 1) to outline the problem as well as make our contributions clearer
- The Background section (Section 2) has been updated to incorporate the literature suggestions from the reviewers.
- We added an experiment that demonstrates the ability of the MPN to adapt to dynamically changing environments at run-time by generating a new low-level controller at each time step (Section 4.3).
- MPN leverages learning from demonstration to acquire the skills necessary to perform the reaching task. For this reason we decided to use a DMP over a simpler proportional controller. This allows us to generate trajectories that reassemble the demonstrated behaviour. An explanation of this feature has been added in Section 4.4.
- Figure 6 (a) and (b) has been added in support of sections 4.3 and 4.4
- Section 4.5 extends the ablations study by evaluating the choices of n-gram sizes as well as the usage of residual layers in the image processing pipeline.
- We added appendix A that further elaborates on the human-subject study and how sentences are generated for our experiments.

---

### Decision · Program_Chairs · 2019-12-19

**Decision:**

Reject

**Comment:**

The present paper addresses the problem of imitation learning in multi-modal settings, combining vision, language and motion. The proposed approach learns an abstract task representation, and the goal is to use this as a basis for generalization. This paper was subject to considerable discussion, and the authors clarified several issues that reviewers raised during the rebuttal phase. Overall, the empirical study presented in the paper remains limited, for example in terms of ablations (which components of the proposed model have what effect on performance) and placement in the context of prior work. As a result, the depth of insights is not yet sufficient for publication.